# Weighted Theta Functions and Embeddings with Applications to Max-Cut, Clustering and Summarization

**Fredrik D. Johansson**
Computer Science & Engineering
Chalmers University of Technology
Göteborg, SE-412 96, Sweden
frejohk@chalmers.se

**Ankani Chattoraj**[*]
Brain & Cognitive Sciences
University of Rochester
Rochester, NY 14627-0268, USA
achattor@ur.rochester.edu

**Chiranjib Bhattacharyya**
Computer Science and Automation
Indian Institute of Science
Bangalore 560012, Karnataka, India
chiru@csa.iisc.ernet.in

**Devdatt Dubhashi**
Computer Science & Engineering
Chalmers University of Technology
Göteborg, SE-412 96, Sweden
dubhashi@chalmers.se

## Abstract

We introduce a unifying generalization of the Lovász theta function, and the associated geometric embedding, for graphs with weights on both nodes and edges. We show how it can be computed exactly by semidefinite programming, and how to approximate it using SVM computations. We show how the theta function can be interpreted as a measure of diversity in graphs and use this idea, and the graph embedding in algorithms for Max-Cut, correlation clustering and document summarization, all of which are well represented as problems on weighted graphs.

## 1 Introduction

Embedding structured data, such as graphs, in geometric spaces, is a central problem in machine learning. In many applications, graphs are attributed with weights on the nodes and edges – information that needs to be well represented by the embedding. Lovász introduced a graph embedding together with the famous theta function in the seminal paper [19], giving his celebrated solution to the problem of computing the Shannon capacity of the pentagon. Indeed, Lovász's embedding is a very elegant and powerful representation of unweighted graphs, that has come to play a central role in information theory, graph theory and combinatorial optimization [10, 8]. However, despite there being at least eight different formulations of $\vartheta(G)$ for unweighted graphs, see for example [20], there does not appear to be a version that applies to graphs with weights on the edges. This is surprising, as it has a natural interpretation in the information theoretic problem of the original definition [19].

A version of the Lovász number for edge-weighted graphs, and a corresponding geometrical representation, could open the way to new approaches to learning problems on data represented as similarity matrices. Here we propose such a generalization for graphs with weights on both nodes and edges, by combining a few key observations. Recently, Jethava et al. [14] discovered an interesting connection between the original theta function and a central problem in machine learning, namely the one class Support Vector Machine (SVM) formulation [14]. This kernel based method gives yet another equivalent characterization of the Lovász number. Crucially, it is easily modified to yield an equivalent characterization of the closely related Delsarte version of the Lovász number

---

[*]This work was performed when the author was affiliated with CSE at Chalmers University of Technology.

introduced by Schrijver [24] which is more flexible and often more convenient to work with. Using this kernel characterization of the Delsarte version of Lovász number, we define a theta function and embedding of weighted graphs, suitable for learning with data represented as similarity matrices.

The original theta function is limited to applications on small graphs, because of its formulation as a semidefinite program (SDP). In [14], Jethava et al. showed that their kernel characterization can be used to compute a number and an embedding of a graph that are often good approximations to the theta function and embedding, and that can be computed fast, scaling to very large graphs. Here we give the analogous approximate method for weighted graphs. We use this approximation to solve the weighted maximum cut problem faster than the classical SDP relaxation.

Finally, we show that our edge-weighted theta function has a natural interpretation as a measure of diversity in graphs. We use this intuition to define a centroid-based correlation clustering algorithm that automatically chooses the number of clusters *and* initializes the centroids. We also show how to use the support vectors, computed in the kernel characterization with both node and edge weights, to perform extractive document summarization.

To summarize our main contributions:

- We introduce a unifying generalization of the famous Lovász number applicable to graphs with weights on both nodes and edges.
- We show that via our characterization, we can compute a good approximation to our weighted theta function and the corresponding embeddings using SVM computations.
- We show that the weighted version of the Lovász number can be interpreted as a measure of diversity in graphs, and we use this to define a correlation clustering algorithm dubbed $\vartheta$-means that automatically a) chooses the number of clusters, and b) initializes centroids.
- We apply the embeddings corresponding to the weighted Lovász numbers to solve weighted maximum cut problems faster than the classical SDP methods, with similar accuracy.
- We apply the weighted kernel characterization of the theta function to document summarization, exploiting both node and edge weights.

## 2   Extensions of Lovász and Delsarte numbers for weighted graphs

**Background**   Consider embeddings of undirected graphs $G = (V, E)$. Lovász introduced an elegant embedding, implicit in the definition of his celebrated theta function $\vartheta(G)$ [19], famously an upper bound on the Shannon capacity and sandwiched between the independence number and the chromatic number of the complement graph.

$$\vartheta(G) = \min_{\{\mathbf{u}_i\}, \mathbf{c}} \max_i \frac{1}{(\mathbf{c}^\top \mathbf{u}_i)^2}, \quad \mathbf{u}_i^\top \mathbf{u}_j = 0, \; \forall (i,j) \notin E, \; \|\mathbf{u}_i\| = \|\mathbf{c}\| = 1 . \tag{1}$$

The vectors $\{\mathbf{u}_i\}$, $\mathbf{c}$ are so-called orthonormal representations or labellings, the dimension of which is determined by the optimization. We refer to both $\{\mathbf{u}_i\}$, and the matrix $U = [\mathbf{u}_1, \ldots, \mathbf{u}_n]$ as an embedding $G$, and use the two notations interchangeably. Jethava et al. [14] introduced a characterization of the Lovász $\vartheta$ function that established a close connection with the one-class support vector machine [23]. They showed that, for an *unweighted* graph $G = (V, E)$,

$$\vartheta(G) \;\; = \;\; \min_{K \in \mathcal{K}(G)} \omega(K), \quad \text{where} \tag{2}$$

$$\mathcal{K}(G) \;\; := \;\; \{K \succeq 0 \mid K_{ii} = 1, K_{ij} = 0, \forall (i,j) \notin E\}, \tag{3}$$

$$\omega(K) \;\; := \;\; \max_{\alpha_i \geq 0} f(\boldsymbol{\alpha}; K), \quad f(\boldsymbol{\alpha}; K) := 2 \sum_i \alpha_i - \sum_{i,j} K_{ij} \alpha_i \alpha_j \tag{4}$$

is the dual formulation of the one-class SVM problem, see [16]. Note that the conditions on $K$ only refer to the non-edges of $G$. In the sequel, $\omega(K)$ and $f(\boldsymbol{\alpha}; K)$ always refer to the definitions in (4).

### 2.1   New weighted versions of $\vartheta(G)$

A key observation in proving (2), is that the set of valid orthonormal representations is equivalent to the set of kernels $\mathcal{K}$. This equivalence can be preserved in a natural way when generalizing the

definition to weighted graphs: any constraint on the inner product $\mathbf{u}_i^T \mathbf{u}_j$ may be represented as constraints on the elements $K_{ij}$ of the kernel matrix.

To define weighted extensions of the theta function, we need to first pass to the closely related Delsarte version of the Lovász number introduced by Schrijver [24]. In the Delsarte version, the orthogonality constraint for non-edges is relaxed to $\mathbf{u}_i^T \mathbf{u}_j \leq 0, (i,j) \notin E$. With reference to the formulation (2) it is easy to observe that the Delsarte version is given by

$$\vartheta^1(G) = \min_{K \in \mathcal{K}^1(G)} \omega(K), \quad \text{where} \quad \mathcal{K}^1(G) := \{K \succeq 0 \mid K_{ii} = 1, K_{ij} \leq 0, \forall (i,j) \notin E\} \quad (5)$$

In other words, the Lovász number corresponds to *orthogonal* labellings of $G$ with orthogonal vectors on the unit sphere assigned to non–adjacent nodes whereas the Delsarte version corresponds to *obtuse* labellings, i.e. the vectors corresponding to non–adjacent nodes are vectors on the unit sphere meeting at obtuse angles. In both cases, the corresponding number is essentially the half-angle of the smallest spherical cap containing all the vectors assigned to the nodes. Comparing (2) and (5) it follows that $\vartheta^1(G) \leq \vartheta(G)$. In the sequel, we will use the Delsarte version and obtuse labellings to define weighted generalizations of the theta function.

We observe in passing, that for any $K \in \mathcal{K}^1$, and for any independent set $I$ in the graph, taking $\alpha_i = 1$ if $i \in I$ and 0 otherwise,

$$\omega(K) \geq 2\sum_i \alpha_i - \sum_{i,j} \alpha_i \alpha_j K_{ij} = \sum_i \alpha_i - \sum_{i \neq j} \alpha_i \alpha_j K_{ij} \geq \sum_i \alpha_i = |I| \quad (6)$$

since for each term in the second sum, either $(i,j)$ is an edge, in which case either $\alpha_i$ or $\alpha_j$ is zero, or $(i,j)$ is a non–edge in which case $K_{ij} \leq 0$. Thus, like $\vartheta(G)$, the Delsarte version $\vartheta^1(G)$ is also an upper bound on the stability or independence number $\alpha(G)$.

**Kernel characterization of theta functions on node-weighted graphs** Lovász number has a classical extension to graphs with node weights $\boldsymbol{\sigma} = [\sigma_1, \ldots, \sigma_n]^\top$, see for example [17]. The generalization, in the Delsarte version (note the inequality constraint), is the following

$$\vartheta(G, \boldsymbol{\sigma}) = \min_{\{\mathbf{u}_i\}, \mathbf{c}} \max_i \frac{\sigma_i}{(\mathbf{c}^\top \mathbf{u}_i)^2}, \quad \mathbf{u}_i^\top \mathbf{u}_j \leq 0, \ \forall (i,j) \notin E, \ \|\mathbf{u}_i\| = \|\mathbf{c}\| = 1 . \quad (7)$$

By passing to the dual of (7), see section 2.1 and [16], we may, as for unweighted graphs, characterize $\vartheta(G, \boldsymbol{\sigma})$ by a minimization over the set of kernels,

$$\mathcal{K}(G, \boldsymbol{\sigma}) := \{K \succeq 0 \mid K_{ii} = 1/\sigma_i, K_{ij} \leq 0, \forall (i,j) \notin E\} \quad (8)$$

and, just like in the unweighted case, $\vartheta^1(G, \boldsymbol{\sigma}) = \min_{K \in \mathcal{K}(G, \boldsymbol{\sigma})} \omega(K)$. When $\sigma_i = 1, \forall i \in V$, this reduces to the unweighted case. We also note that for any $K \in \mathcal{K}(G, \boldsymbol{\sigma})$ and for any independent set $I$ in the graph, taking $\alpha_i = \sigma_i$ if $i \in I$ and 0 otherwise,

$$\omega(K) \geq 2\sum_i \alpha_i - \sum_{i,j} \alpha_i \alpha_j K_{ij} = 2\sum_{i \in I} \sigma_i - \sum_{i \in I} \frac{\sigma_i^2}{\sigma_i} - \sum_{i \neq j} \alpha_i \alpha_j K_{ij} \geq \sum_{i \in I} \sigma_i , \quad (9)$$

since $K_{ij} \leq 0 \ \forall (i,j) \notin E$. Thus, $\vartheta^1(G, \boldsymbol{\sigma}) \geq \omega(K)$ is an upper bound on the weight of the maximum-weight independent set.

**Extension to edge-weighted graphs** The kernel characterization of $\vartheta^1(G)$ allows one to define a natural extension to data given as similarity matrices represented in the form of a weighted graph $G = (V, S)$. Here, $S$ is a similarity function on (unordered) node pairs, and $S(i,j) \in [0,1]$ with $+1$ representing complete similarity and 0 complete dissimilarity. The obtuse labellings corresponding to the Delsarte version are somewhat more flexible even for unweighted graphs, but is particularly well suited for weighted graphs. We define

$$\vartheta^1(G, S) := \min_{K \in \mathcal{K}(G, S)} \omega(K) \quad \text{where} \quad \mathcal{K}(G, S) := \{K \succeq 0 \mid K_{ii} = 1, K_{ij} \leq S_{ij}\} \quad (10)$$

In the case of an unweighted graph, where $S_{ij} \in \{0, 1\}$, this reduces exactly to (5).

Table 1: Characterizations of weighted theta functions. In the first row are characterizations following the original definition. In the second are kernel characterizations. The bottom row are versions of the LS-labelling [14]. In all cases, $\|\mathbf{u}_i\| = \|\mathbf{c}\| = 1$. $A$ refers to the adjacency matrix of $G$.

| Unweighted | Node-weighted | Edge-weighted |
|---|---|---|
| $\min\limits_{\{\mathbf{u}_i\}} \min\limits_{\mathbf{c}} \max\limits_{i} \dfrac{1}{(\mathbf{c}^\top \mathbf{u}_i)^2}$ $\mathbf{u}_i^\top \mathbf{u}_j \leq 0, \ \forall (i,j) \notin E$ | $\min\limits_{\{\mathbf{u}_i\}} \min\limits_{\mathbf{c}} \max\limits_{i} \dfrac{\sigma_i}{(\mathbf{c}^\top \mathbf{u}_i)^2}$ $\mathbf{u}_i^\top \mathbf{u}_j = 0, \ \forall (i,j) \notin E$ | $\min\limits_{\{\mathbf{u}_i\}} \min\limits_{\mathbf{c}} \max\limits_{i} \dfrac{1}{(\mathbf{c}^\top \mathbf{u}_i)^2}$ $\mathbf{u}_i^\top \mathbf{u}_j \leq S_{ij}, \ i \neq j$ |
| $\mathcal{K}_G = \{K \succeq 0 \mid K_{ii} = 1,$ $K_{ij} = 0, \forall (i,j) \notin E\}$ | $\mathcal{K}_{G,\boldsymbol{\sigma}} = \{K \succeq 0 \mid K_{ii} = 1/\sigma_i,$ $K_{ij} = 0, \forall (i,j) \notin E\}$ | $\mathcal{K}_{G,S} = \{K \succeq 0 \mid K_{ii} = 1,$ $K_{ij} \leq S_{ij}, i \neq j\}$ |
| $K_{LS} = \dfrac{A}{\|\lambda_n(A)\|} + I$ | $K_{LS}^{\boldsymbol{\sigma}} = \dfrac{A}{\sigma_{max}\|\lambda_n(A)\|} + \text{diag}(\boldsymbol{\sigma})^{-1}$ | $K_{LS}^S = \dfrac{S}{\|\lambda_n(S)\|} + I$ |

**Unifying weighted generalization**   We may now combine both node and edge weights to form a fully general extension to the Delsarte version of the Lovász number,

$$\vartheta^1(G, \boldsymbol{\sigma}, S) = \min_{K \in \mathcal{K}(G, \boldsymbol{\sigma}, S)} \omega(K), \quad \mathcal{K}(G, \boldsymbol{\sigma}, S) := \left\{ K \succeq 0 \mid K_{ii} = \frac{1}{\sigma_i}, K_{ij} \leq \frac{S_{ij}}{\sqrt{\sigma_i \sigma_j}} \right\} \quad (11)$$

It is easy to see that for unweighted graphs, $S_{ij} \in \{0, 1\}$, $\sigma_i = 1$, the definition reduces to the Delsarte version of the theta function in (5). $\vartheta^1(G, \boldsymbol{\sigma}, S)$ is hence a strict generalization of $\vartheta^1(G)$. *All the proposed weighted extensions are defined by the same objective, $\omega(K)$. The only difference is the set $\mathcal{K}$, specialized in various ways, over which the minimum, $\min_{K \in \mathcal{K}} \omega(K)$, is computed.* It also is important to note, that with the generalization of the theta function comes an implicit generalization of the geometric representation of $G$. Specifically, for any feasible $K$ in (11), there is an embedding $U = [\mathbf{u}_1, \ldots, \mathbf{u}_n]$ such that $K = U^\top U$ with the properties $\mathbf{u}_i^\top \mathbf{u}_j \sqrt{\sigma_i \sigma_j} \leq S_{ij}$, $\|\mathbf{u}_i\|_2 = 1/\sqrt{\sigma_i}$, which can be retrieved using matrix decomposition. Note that $\mathbf{u}_i^\top \mathbf{u}_j \sqrt{\sigma_i \sigma_j}$ is exactly the cosine similarity between $\mathbf{u}_i$ and $\mathbf{u}_j$, which is a very natural choice when $S_{ij} \in [0, 1]$.

The original definition of the (Delsarte) theta function and its extensions, as well as their kernel characterizations, can be seen in table 1. We can prove the equivalence of the embedding (top) and kernel characterizations (middle) using the following result.

**Proposition 2.1.** *For any embedding $U \in \mathbb{R}^{d \times n}$ with $K = U^\top U$, and $f$ in (4), the following holds*

$$\min_{\mathbf{c} \in \mathcal{S}^{d-1}} \max_i \frac{1}{(\mathbf{c}^\top \mathbf{u}_i)^2} = \max_{\alpha_i \geq 0} f(\boldsymbol{\alpha}; K) \ . \quad (12)$$

*Proof.* The result is given as part of the proof of Theorem 3 in Jethava et al. [14]. See also [16]. ∎

As we have already established in section 2 that any set of geometric embeddings have a characterization as a set of kernel matrices, it follows that the minimizing the LHS in (12) over a (constrained) set of orthogonal representations, $\{\mathbf{u}_i\}$, is equivalent to minimizing the RHS over a kernel set $\mathcal{K}$.

## 3   Computation and fixed-kernel approximation

The weighted generalization of the theta function, $\vartheta^1(G, \boldsymbol{\sigma}, S)$, defined in the previous section, may be computed as a semidefinite program. In fact $\vartheta^1(G, \boldsymbol{\sigma}, S) = 1/(t^*)^2$ for $t^*$, the solution to the following problem. For details, see the supplementary material [16].

$$\begin{aligned}
\underset{X}{\text{minimize}} \quad & t \quad \text{subject to} \quad X \succeq 0, \ X \in \mathbb{R}^{(n+1) \times (n+1)} \\
& X_{i,n+1} \geq t, \ X_{ii} = 1/\sigma_i, \quad i \in [n] \\
& X_{ij} \leq S_{ij}/\sqrt{\sigma_i \sigma_j}, \quad\quad\quad i \neq j, \ i, j \in [n]
\end{aligned} \quad (13)$$

While polynomial in time complexity [13], solving the SDP is too slow in many cases. To address this, Jethava et al. [14] introduced a fast approximation to (the unweighted) $\vartheta(G)$, dubbed SVM-theta. They showed that in some cases, the minimization over $\mathcal{K}$ in (2) can be replaced by a fixed choice of $K$, while causing just a constant-factor error. Specifically, for unweighted graphs with adjacency matrix $A$, Jethava et al. [14] defined the so called LS-labelling, $K_{LS}(G) = A/|\lambda_n(A)| + I$, and showed that for large families of graphs $\vartheta(G) \leq \omega(K_{LS}(G)) \leq \gamma\vartheta(G)$, for a constant $\gamma$.

We extend the LS-labelling to weighted graphs. For graphs with edge weights, represented by a similarity matrix $S$, the original definition may be used, with $S$ substituted for $A$. For node weighted graphs we also must satisfy the constraint $K_{ii} = 1/\sigma_i$, see (8). A natural choice, still ensuring positive semidefiniteness is,

$$K_{LS}(G, \boldsymbol{\sigma}) = \frac{A}{\sigma_{max}|\lambda_n(A)|} + \text{diag}(\boldsymbol{\sigma})^{-1} \tag{14}$$

where $\text{diag}(\boldsymbol{\sigma})^{-1}$ is the diagonal matrix $\Sigma$ with elements $\Sigma_{ii} = 1/\sigma_i$, and $\sigma_{max} = \max_{i=1}^n \sigma_i$. Both weighted versions of the LS-labelling are presented in table 1. The fully generalized labelling, for graphs with weights on both nodes and edges, $K_{LS}(G, \boldsymbol{\sigma}, S)$ can be obtained by substituting $S$ for $A$ in (14). As with the exact characterization, we note that $K_{LS}(G, \boldsymbol{\sigma}, S)$ reduces to $K_{LS}(G)$ for the uniform case, $S_{ij} \in \{0, 1\}$, $\sigma_i = 1$. For all versions of the LS-labelling of $G$, as with the exact characterization, a geometric embedding $U$ may be obtained from $K_{LS}$ using matrix decompotion.

## 3.1 Computational complexity

Solving the full problem in the kernel characterization (11), is not faster than the computing the SDP characterization (13). However, for a fixed $K$, the one-class SVM can be solved in $O(n^2)$ time [12]. Retrieving the embedding $U : K = U^T U$ may be done using Cholesky or singular value decomposition (SVD). In general, algorithms for these problems have complexity $O(n^3)$. However, in many cases, a rank $d$ approximation to the decomposition is sufficient, see for example [9]. A thin (or truncated) SVD corresponding to the top $d$ singular values may be computed in $O(n^2 d)$ time [5] for $d = O(\sqrt{n})$. The remaining issue is the computation of $K$. The complexity of computing the LS-labelling discussed in the previous section is dominated by the computation of the minimum eigenvalue $\lambda_n(A)$. This can be done approximately in $\tilde{O}(m)$ time, where $m$ is the number of edges of the graph [1]. Overall, the complexity of computing both the embedding $U$ and $\omega(K)$ is $O(dn^2)$.

## 4 The theta function as diversity in graphs: $\vartheta$-means clustering

In section 2, we defined extensions of the Delsarte version of the Lovász number, $\vartheta^1(G)$ and the associated geometric embedding, for weighted graphs. Now we wish to show how both $\vartheta(G)$ and the geometric embedding are useful for solving common machine learning tasks. We build on an intuition of $\vartheta(G)$ as a measure of *diversity* in graphs, illustrated here by a few simple examples. For complete graphs $K_n$, it is well known that $\vartheta(K_n) = 1$, and for empty graphs $\overline{K}_n$, $\vartheta(\overline{K}_n) = n$. We may interpret these graphs as having $1$ and $n$ clusters respectively. Graphs with several disjoint clusters make a natural middle-ground. For a graph $G$ that is a union of $k$ disjoint cliques, $\vartheta(G) = k$.

Now, consider the analogue of (6) for graphs with edge weights $S_{ij}$. For any $K \in \mathcal{K}(G, S)$ and for any subset $H$ of nodes, let $\alpha_i = 1$ if $i \in H$ and 0 otherwise. Then, since $K_{ij} \leq S_{ij}$,

$$2\sum_i \alpha_i - \sum_{ij} \alpha_i\alpha_j K_{ij} \quad = \quad \sum_i \alpha_i - \sum_{i \neq j} \alpha_i\alpha_j K_{ij} \geq |H| - \sum_{i \neq j, i,j \in H} S_{ij} \,.$$

Maximizing this expression may be viewed as the trade-off of finding a subset of nodes that is both *large* and *diverse*; *the objective function is the size of the set subjected to a penalty for non–diversity.*

In general support vector machines, non-zero support values $\alpha_i$ correspond to support vectors, defining the decision boundary. As a result, *nodes $i \in V$ with high values $\alpha_i$ may be interpreted as an important and diverse set of nodes.*

## 4.1 $\vartheta$-means clustering

A common problem related to diversity in graphs is correlation clustering [3]. In correlation clustering, the task is to cluster a set of items $V = \{1, \ldots, n\}$, based on their similarity, or correlation,

---
**Algorithm 1** $\vartheta$-means clustering
---
1: **Input:** Graph $G$, with weight matrix $S$ and node weights $\boldsymbol{\sigma}$.

2: Compute kernel $K \in \mathcal{K}(G, \boldsymbol{\sigma}, S)$

3: $\alpha_i^* \leftarrow \arg\max_{\alpha_i} f(\alpha; K)$, as in (4)

4: Sort alphas according to $j_i$ such that $\alpha_{j_1} \geq \alpha_{j_2} \geq ... \geq \alpha_{j_n}$

5: Let $k = \lceil \hat{\vartheta} \rceil$ where $\hat{\vartheta} \leftarrow \omega(K) = f(\alpha^*; K)$

6: **either a)**

7:    Initialize labels $Z_i = \arg\max_{j \in \{j_1, ..., j_k\}} K_{ij}$

8:   **Output:** result of kernel $k$-means with kernel $K$, $k = \lceil \hat{\vartheta} \rceil$ and $Z$ as initial labels

9: **or b)**

10:    Compute $U : K = U^T U$, with columns $U_i$, and let $\mathcal{C} \leftarrow \{U_{j_i} : i \leq k\}$

11:   **Output:** result of $k$-means with $k = \lceil \hat{\vartheta} \rceil$ and $\mathcal{C}$ as initial cluster centroids
---

$S : V \times V \to \mathbb{R}^{n \times n}$, *without* specifying the number of clusters beforehand. This is naturally posed as a problem of clustering the nodes of an edge-weighted graph. In a variant called *overlapping* correlation clustering [4], items may belong to several, overlapping, clusters. The usual formulation of correlation clustering is an integer linear program [3]. Making use of geometric embeddings, we may convert the graph clustering problem to the more standard problem of clustering a set of points $\{\mathbf{u}_i\}_{i=1}^n \in \mathbb{R}^{d \times n}$, allowing the use of an arsenal of established techniques, such as $k$-means clustering. However, we remind ourselves of two common problems with existing clustering algorithms.

**Problem 1: Number of clusters** Many clustering algorithms relies on the user making a good choice of $k$, the number of clusters. As this choice can have dramatic effect on both the accuracy and speed of the algorithm, heuristics for choosing $k$, such as Pham et al. [22], have been proposed.

**Problem 2: Initialization** Popular clustering algorithms such as Lloyd's $k$-means, or expectation-maximization for Gaussian mixture models require an initial guess of the parameters. As a result, these algorithms are often run repeatedly with different random initializations.

We propose solutions to both problems based on $\vartheta^1(G)$. To solve Problem 1, we choose $k = \lceil \vartheta^1(G) \rceil$. This is motivated by $\vartheta^1(G)$ being a measure of diversity. For Problem 2, we propose initializing parameters based on the observation that the non-zero $\alpha_i$ are support vectors. Specifically, we let the initial clusters by represented by the set of $k$ nodes, $I \subset V$, with the largest $\alpha_i$. In $k$-means clustering, this corresponds to letting the initial centroids be $\{\mathbf{u}_i\}_{i \in I}$. We summarize these ideas in algorithm 1, comprising both $\vartheta$-means and kernel $\vartheta$-means clustering.

In section 3.1, we showed that computating the approximate weighted theta function and embedding, can be done in $O(dn^2)$ time for a rank $d = O(\sqrt{n})$ approximation to the SVD. As is well-known, Lloyd's algorithm has a very high worst-case complexity and will dominate the overall complexity.

# 5 Experiments

## 5.1 Weighted Maximum Cut

The maximum cut problem (Max-Cut), a fundamental problem in graph algorithms, with applications in machine learning [25], has famously been solved using geometric embeddings defined by semidefinite programs [9]. Here, given a graph $G$, we compute an embedding $U \in \mathbb{R}^{d \times n}$, the SVM-theta labelling in [15], using the LS-labelling, $K_{LS}$. To reduce complexity, while preserving accuracy [9], we use a rank $d = \sqrt{2n}$ truncated SVD, see section 3.1. We apply the Goemans-Williamson random hyperplane rounding [9] to partition the embedding into two sets of points, representing the cut. The rounding was repeated 5000 times, and the maximum cut is reported.

Helmberg & Rendl [11] constructed a set of 54 graphs, 24 of which are weighted, that has since often been used as benchmarks for Max-Cut. We use the six of the *weighted* graphs for which there are multiple published results [6, 21]. Our approach is closest to that of the SDP-relaxation, which

Table 2: Weighted maximum cut. $c$ is the weight of the produced cut.

| | SDP [6] | | SVM-$\vartheta$ | | Best known [21] | |
| --- | --- | --- | --- | --- | --- | --- |
| **Graph** | $c$ | **Time** | $c$ | **Time** | $c$ | **Time** |
| G11 | 528 | 165s | 522 | 3.13s | 564 | 171.8s |
| G12 | 522 | 145s | 518 | 2.94s | 556 | 241.5s |
| G13 | 542 | 145s | 540 | 2.97s | 580 | 227.5s |
| G32 | 1280 | 1318s | 1286 | 35.5s | 1398 | 900.6s |
| G33 | 1248 | 1417s | 1260 | 36.4s | 1376 | 925.6s |
| G34 | 1264 | 1295s | 1268 | 37.9s | 1372 | 925.6s |

Table 3: Clustering of the (mini) newsgroup dataset. Average (and std. deviation) over 5 splits. $\hat{k}$ is the average number of clusters predicted. The true number is $k = 16$.

| | $\mathbf{F}_1$ | $\hat{k}$ | **Time** |
| --- | --- | --- | --- |
| VOTE/BOEM | $31.29 \pm 4.0$ | 124 | 8.7m |
| PIVOT/BOEM | $30.07 \pm 3.4$ | 120 | 14m |
| BEST/BOEM | $29.67 \pm 3.4$ | 112 | 13m |
| FIRST/BOEM | $26.76 \pm 3.8$ | 109 | 14m |
| $k$-MEANS+RAND | $17.31 \pm 1.3$ | 2 | 15m |
| $k$-MEANS+INIT | $20.06 \pm 6.8$ | 3 | 5.2m |
| $\vartheta$-MEANS+RAND | $35.60 \pm 4.3$ | 25 | 45s |
| $\vartheta$-MEANS | $36.20 \pm 4.9$ | 25 | 11s |

has time complexity $O(mn \log^2 n/\epsilon^3)$ [2]. In comparison, our method takes $O(n^{2.5})$ time, see section 3.1. The results are presented in table 2. For all graphs, the SVM approximation is comparable to or better than the SDP solution, and considerably faster than the best known method [21].[1]

## 5.2 Correlation clustering

We evaluate several different versions of algorithm 1 in the task of correlation clustering, see section 4.1. We consider a) the full version ($\vartheta$-MEANS), b) one with $k = \lceil \hat{\vartheta} \rceil$ but random initialization of centroids ($\vartheta$-MEANS+RAND), c) one with $\alpha$-based initialization but choosing $k$ according to Pham et al. [22] ($k$-MEANS+INIT) and d) $k$ according to [22] and random initialization ($k$-MEANS+RAND). For the randomly initialized versions, we use 5 restarts of $k$-means++. In all versions, we cluster the points of the embedding defined by the fixed kernel (LS-labelling) $K = K_{LS}(G, S)$.

Elsner & Schudy [7] constructed five affinity matrices for a subset of the classical 20-newsgroups dataset. Each matrix, corresponding to a different split of the data, represents the similarity between messages in 16 different newsgroups. The task is to cluster the messages by their respective newsgroup. We run algorithm 1 on every split, and compute the $F_1$-score [7], reporting the average and standard deviation over all splits, as well as the predicted number of clusters, $\hat{k}$. We compare our results to several greedy methods described by Elsner & Schudy [7], see table 3. We only compare to their logarithmic weighting schema, as the difference to using additive weights was negligible [7].

The results are presented in table 3. We observe that the full $\vartheta$-means method achieves the highest $F_1$-score, followed by the version with random initialization (instead of using embeddings of nodes with highest $\alpha_i$, see algorithm 1). We note also that choosing $k$ by the method of Pham et al. [22] consistently results in too few clusters, and with the greedy search methods, far too many.

## 5.3 Overlapping Correlation Clustering

Bonchi et al. [4] constructed a benchmark for *overlapping* correlation clustering based on two datasets for multi-label classification, Yeast and Emotion. The datasets consist of 2417 and 593 items belonging to one or more of 14 and 6 overlapping clusters respectively. Each set can be represented as an $n \times k$ binary matrix $L$, where $k$ is the number of clusters and $n$ is the number of items,

Table 4: Clustering of the Yeast and Emotion datasets. [†]The total time for finding the best solution. The times for OCC-ISECT *for a single $k$* was 2.21s and 80.4s respectively.

| | Emotion | | | | Yeast | | | |
| --- | --- | --- | --- | --- | --- | --- | --- | --- |
| | Prec. | Rec. | $F_1$ | Time | Prec. | Rec. | $F_1$ | Time |
| OCC-ISECT [4] | 0.98 | 1 | 0.99 | 12.1[†] | 0.99 | 1.00 | 1.00 | 716s[†] |
| $\vartheta$-means (no $k$-means) | 1 | 1 | 1 | 0.34s | 0.94 | 1 | 0.97 | 6.67s |

such that $L_{ic} = 1$ iff item $i$ belongs to cluster $c$. From $L$, a weight matrix $S$ is defined such that $S_{ij}$ is the Jaccard coefficient between rows $i$ and $j$ of $L$. $S$ is often sparse, as many of the pairs do not share a single cluster. The correlation clustering task is to reconstruct $L$ from $S$.

Here, we use *only* the centroids $\mathcal{C} = \{\mathbf{u}_{j_1}, ..., \mathbf{u}_{j_k}\}$ produced by algorithm 1, without running $k$-means. We let each centroid $c = 1, ..., k$ represent a cluster, and assign a node $i \in V$ to that cluster, i.e. $\hat{L}_{ic} = 1$, iff $\mathbf{u}_i^T \mathbf{u}_{j_c} > 0$. We compute the precision and recall following Bonchi et al. [4]. For comparison with Bonchi et al. [4], we run their algorithm called OCC-ISECT with the parameter $\bar{k}$, bounding the number of clusters, in the interval $1, ..., 16$ and select the one resulting in lowest cost.

The results are presented in table 4. For Emotion and Yeast, $\vartheta$-means estimated the number of clusters, $k$ to be 6 (the correct number) and 8 respectively. For OCC-Isect, the $k$ with the lowest cost were 10 and 13. We note that while very similar in performance, the $\vartheta$-means algorithms is considerably faster than OCC-ISECT, especially when $k$ is unknown.

## 5.4 Document summarization

Finally, we briefly examine the idea of using $\alpha_i$ to select a both relevant and diverse set of items, in a very natural application of the weighted theta function – extractive summarization [18]. In extractive summarization, the goal is to automatically summarize a text by picking out a small set of sentences that best represents the whole text. We may view the sentences of a text as the nodes of a graph, with edge weights $S_{ij}$, the similarity between sentences, and node weights $\sigma_i$ representing the relevance of the sentence to the text as a whole. The trade-off between brevity and relevance described above can then be viewed as finding a set of nodes that has both high total weight and high diversity. This is naturally accomplished using our framework by computing $[\alpha_1^*, \dots, \alpha_n^*]^\top = \arg\max_{\alpha_i > 0} f(\boldsymbol{\alpha}; K)$ for fixed $K = K_{LS}(G, \boldsymbol{\sigma}, S)$ and picking the sentences with the highest $\alpha_i^*$.

We apply this method to the multi-document summarization task of DUC-04[2]. We let $S_{ij}$ be the TF-IDF sentence similarity described by Lin & Bilmes [18], and let $\sigma_i = (\sum_j S_{ij})^2$. State-of-the-art systems, purpose-built for summarization, achieve around 0.39 in recall and $F_1$ score [18]. Our method achieves a score of 0.33 on both measures which is about the same as the basic version of [18]. This is likely possible to improve by tuning the trade-off between relevance and diversity, such as a making a more sophisticated choice of $S$ and $\boldsymbol{\sigma}$. However, we leave this to future work.

## 6 Conclusions

We have introduced a unifying generalization of Lovász's theta function and the corresponding geometric embedding to graphs with node and edge weights, characterized as a minimization over a constrained set of kernel matrices. This allows an extension of a fast approximation of the Lovász number to weighted graphs, defined by an SVM problem for a fixed kernel matrix. We have shown that the theta function has a natural interpretation as a measure of diversity in graphs, a useful function in several machine learning problems. Exploiting these results, we have defined algorithms for weighted maximum cut, correlation clustering and document summarization.

**Acknowledgments**

This work is supported in part by the Swedish Foundation for Strategic Research (SSF).

## Footnotes

[1]Note that the timing results for the SDP method are from the original paper, published in 2001.

[2]http://duc.nist.gov/duc2004/

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
