[Supplementary Material]

# Supplementary material for Weighted Theta Functions and Embeddings with Applications to Max-Cut, Clustering and Summarization

**Fredrik D. Johansson**
Computer Science & Engineering
Chalmers University of Technology
Göteborg, SE-412 96, Sweden
frejohk@chalmers.se

**Ankani Chattoraj**[*]
Brain & Cognitive Sciences
University of Rochester
Rochester, NY 14627-0268, USA
achattor@ur.rochester.edu

**Chiranjib Bhattacharyya**
Computer Science and Automation
Indian Institute of Science
Bangalore 560012, Karnataka, India
chiru@csa.iisc.ernet.in

**Devdatt Dubhashi**
Computer Science & Engineering
Chalmers University of Technology
Göteborg, SE-412 96, Sweden
dubhashi@chalmers.se

## 1 Equivalence of kernel and embedding characterizations

We wish to show that for any $U$ with $K = U^T U$, we have

$$\min_{\mathbf{c} \in \mathcal{S}^{d-1}} \max_i \frac{1}{(\mathbf{c}^\top \mathbf{u}_i)^2} = \max_{\alpha_i \geq 0} f(\boldsymbol{\alpha}; K) \ .$$

We observe that the LHS is equivalent (with unit vector constraints implicit) to

$$\min_{\mathbf{c}, t} t^2 \quad \text{subject to} \quad \mathbf{u}_i^\top \mathbf{c} \geq \frac{1}{t}$$

Now, let $\mathbf{w} = 2t\mathbf{c}$, and note $t^2 = \|\mathbf{w}\|^2/4$. Further, $\mathbf{c}^\top \mathbf{u}_i = \mathbf{w}^\top \mathbf{u}_i/(2t)$. Hence, the original problem is equivalent to

$$\min_{\mathbf{w}} \frac{\|\mathbf{w}\|^2}{4} \quad \text{subject to} \quad \mathbf{w}^\top \mathbf{u}_i \geq 2$$

We form the Lagrange dual. The Lagrangian is then,

$$L(\mathbf{w}, \alpha) = \frac{\|\mathbf{w}\|^2}{4} + \sum_{i=1}^n \alpha_i (2 - \mathbf{w}^\top \mathbf{u}_i)$$

We set the gradient to zero.

$$\nabla_{\mathbf{w}} L = \frac{\mathbf{w}}{2} - \sum_{i=1}^n \alpha_i \mathbf{u}_i = \mathbf{0}$$

This gives $\mathbf{w} = 2\sum_{i=1}^n \alpha_i \mathbf{u}_i$. Hence, the dual problem is a maximization over

$$\|2\sum_{i=1}^n \alpha_i \mathbf{u}_i\|^2 + \sum_{i=1}^n \alpha_i (2 - 2\sum_{j=1}^n \alpha_j \mathbf{u}_i^\top \mathbf{u}_j)$$

---

[*]This work was performed when the author was affiliated with Chalmers University of Technology.

which can be rewritten as, with $K = U^\top U$,

$$2 \sum_{i=1}^{n} \alpha_i - \sum_{i,j=1}^{n} \alpha_i \alpha_j K_{ij} := f(\boldsymbol{\alpha}; K)$$

Note that this argument extends to the node-weighted version, either by a variable substitution $\mathbf{u}_i' = \mathbf{u}_i/\sqrt{\sigma_i}$ or by a simple modification of the derivation.

## 2    Formulation as semidefinite program

Consider the kernel characterization of the fully weighted theta function

$$\vartheta^1(G, \boldsymbol{\sigma}, S) = \min_{K \in \mathcal{K}(G, \boldsymbol{\sigma}, S)} \omega(K), \quad \mathcal{K}(G, \boldsymbol{\sigma}, S) := \left\{ K \succeq 0 \mid K_{ii} = \frac{1}{\sigma_i}, K_{ij} \leq \frac{S_{ij}}{\sqrt{\sigma_i \sigma_j}} \right\}$$

This can, by the results in the previous section, be written as an on optimization problem over a set of orthogonal representations,

$$\vartheta(G, \boldsymbol{\sigma}) = \min_{\{\mathbf{u}_i\}, \mathbf{c}} \max_i \frac{\sigma_i}{(\mathbf{c}^\top \mathbf{u}_i)^2}, \quad \mathbf{u}_i^\top \mathbf{u}_j \leq S_{ij}, \quad \|\mathbf{u}_i\| = \|\mathbf{c}\| = 1 \, .$$

Similar to the previous section, we may rewrite the above problem as

$$\frac{1}{\sqrt{\vartheta(G, \boldsymbol{\sigma})}} = \max_{\{\mathbf{u}_i\}, \mathbf{c}} t, \quad \frac{\mathbf{c}^\top \mathbf{u}_i}{\sqrt{\sigma_i}} \geq t, \quad \mathbf{u}_i^\top \mathbf{u}_j \leq S_{ij}, \quad \|\mathbf{u}_i\| = \|\mathbf{c}\| = 1 \, .$$

Now, consider the matrix $(n+1) \times (n+1)$ matrix $X$ where

$$X_{ij} = \begin{cases} \frac{\mathbf{u}_i^\top \mathbf{u}_j}{\sqrt{\sigma_i \sigma_j}}, & i \neq j, i, j \leq n \\ \frac{\mathbf{c}^\top \mathbf{u}_i}{\sqrt{\sigma_i}}, & i \leq n, j = n+1 \text{ or } j \leq n, i = n+1 \\ \frac{1}{\sigma_i}, & i = j, i \leq n \\ 1, & i = j = n+1 \end{cases}$$

It is easy to see that $X = U^\top U$ where $U$ is a matrix with columns $[\mathbf{u}_1/\sqrt{\sigma_1}, ..., \mathbf{u}_n/\sqrt{\sigma_n}, \mathbf{c}]$. Consequently, $X$ is positive semidefinite. Now, it is also plain to see that the optimization in (13) is equivalent to the one above.