[Reviews · NeurIPS 2015]

Submitted by Assigned_Reviewer_1

The paper starts by situating the problem and motivating their approach - essentially, enabling embeddings for weighted graphs by extending the original implicit embedding performed in Lovasz "Shannon capacity of a graph" paper, which operated on binary graphs. The paper then presents the connection between kernel machines and the Lovasz number in the unweighted case, and extends previous work for vertex-weighted, edge-weighted, and LS-labelled graphs. Section 3 provides practical details for computation, and section 4 motivates the use of their approach for the clustering problem - setting the number of clusters by using their \vartheta^1 bound, and initialising the clusters by starting by vertices with large alpha_i values. Finally, they show results on max-cut, clustering, overlapping clustering, and summarization tasks.

The paper ties together very different work to propose a coherent approach to graph embedding. The contributions are clearly laid out, and the references to previous work is well established and used.

Overall, the paper is well-written and the notation is clean.

and the derivation of their technique is clearly explained.

This is significant work that can have impact in several domains of machine learning, as shown in the experiments section.

There are a few issues that deserve slightly better explanations:

a) To recover the embedding from the computed kernel matrix, the authors propose to use a truncated SVD. This is fine, but the efficiency of the embedding really hinges on the choice of the truncation order. How much better than a full-rank orthonormal representation (basically order n) can we do and still have good performance? Surely, this depends on the structure of the input adjacency or affinity matrix - something that could be generated from a stochastic block model, for example, would presumably call for a different choice of truncation order than a lattice graph? The authors choose \sqrt(n) for the truncation order, but it would be useful to have some intuition as to the impact on the quality of the embedding generated (the paper focuses a lot on how to obtain K and alpha).

b) In section 4, the authors suggest using vertices with the largest alphas as initial centroids. In the SVM analogy, these would be support vectors (i.e. non-zero alphas), which would put them on the decision boundary. However, in the SVM setting these points are typically outliers or extreme values and don't represent the bulk of the data. I understand that the authors want 'diversity' here, but the explanation in lines 263-265 seems to be at odds with the stated goal of finding good centroids in section 4.1 - do we want representative points (center of mass) or influential points (extremes, on the boundary in the discriminative setting) ?

c) In the experiments on maxcut (table 2), the authors have selected 6 graphs from the Helmberg and Rendl set of 24 weighted graphs. These are toroidal graphs, which have a known upper bound on the cut size. Because the comparison to the best known algorithm (Scatter search or VNSPR) if empirical anyways, is there a particular reason to select these? How would the method perform on random graphs (e.g. G6...G10) or planar graphs (e.g. G18...G21) ?
Summary: This is a nice contribution with wide applicability. The paper is clearly written and the example applications are convincing.

Submitted by Assigned_Reviewer_2

In this paper the authors extend the graph embedding framework based on Lovasz theta function.

The proposed graph embedding exploits the capability of the Support Vector Machine to reduce the computational complexity of the embedding. The extension can handle those cases where weights are assigned either to the nodes or to the edges. The proposed algorithm is then used for clustering the nodes with respect to the structure of the graph. The objective of the paper is well defined and the method introduced technically and theoretically sounds.

As far as I know the interpretation of Lovasz theta function as diversity is novel approach to graph embedding and node clustering.

Summary: The recognition and exploitation that the theta function can measure the structural diversity of a graph allows the authors to introduce a new clustering algorithm built on the Support Vector Machine. That algorithm is significantly faster than the known Semidefinite programming based approach.

Submitted by Assigned_Reviewer_3

DEFINITION OF THETA FUNCTIONS ON NODE- and EDGE-WEIGHTED GRAPHS

A theta function is defined for node-weighted graphs by replacing the "1" in the numerator of the quotient defining the theta function by the corresponding node weight. What does this mean in terms of modeling? What is the consequence of the node weight for the corresponding embedding? Why is this a reasonable extension? What happens for a node with weight 0 (or very close to 0), and is this reasonable?

Similarly, what is the impact of the extension to edge-weighted graphs in (10)? What does this do to the corresponding embedding? In fact, it is not clear to me exactly what values the kernel K is allowed to take -- from (3) I thought it was binary, but in (10) it is clearly not.

STYLE

The paper is unfortunately written in a somewhat sloppy style, which makes it hard to follow. I understand that space is very tight, but some notation is so key (and easy to specify) that it should be possible to squeeze in:

* The definition of E with respect to the graph G * The dimensionality of the space in which the u_i live * The definition of the K appearing from equation (2) * In that respect: If you wrote "K_{ij} = 0 \forall (i,j) \notin E" with a "\forall" in place of the comma, this would already ease readability, and there are many similar points where I have to read a couple of times before deciding what you most likely meant.

These are just examples from the introduction where I was not sure when reading it, what you meant. Even for someone well familiar with graph mining but not intimately familiar with Lovasz theta functions (=me), it would be nice if the introductory definitions could serve as a reminder -- otherwise, why are they there?

EXPERIMENTS

Section 5.1. Why do you only present results on 6 out of 24 weighted graphs? How do I know that you did not just cherry-pick the best results? Why do you not compare to the original theta function without weighted nodes or edges, but with a binarization depending on the weights? How do I know that including the weights did, in fact, contribute to the solution of the problem?
Summary: SUMMARY

This paper extends the Lovasz theta function on graphs and its corresponding geometric embedding framework to graphs with real-valued weights on nodes and edges. This is used to generate methods for max-cut, correlation clustering, overlapping correlation clustering and document summarization. Results are not amazing but there seem to be significant runtime advantages. There is no comparison with the unweighted version for binarized graphs. Unfortunately the paper is written in such a superficial way that it is not clear what the method does and what its implications are. The paper might have valuable contributions but seems a bit premature.

Submitted by Assigned_Reviewer_4

The authors first present the Lovasz and Delsarte numbers for graphs.

Then they look at a recent method of computing the Lovasz number via a one-class SVM problem.

They then extend to both edge and node weighted versions via some very appealing intuitive equations (6)-(11) (summarized in a table).

As an approximation method, they employ an SVM optimization using a preselected kernel (LS-labeling).

This method has low complexity and reasonable approximation to the desired quantity.

The authors then apply their methods to weighted maximum cut and also to clustering.

Both methods yield interesting results and appear promising.

Overall, this paper is a good balance between theory and application.

Using the kernel upper bound as an optimization and proof method fits well with machine learning approaches.

In addition, the good performance shows significant promise of the methods.

Summary: The authors look at generalizing the Lovasz theta to weighted graphs and corresponding applications.

The paper presents a nice balance of theoretical results and practical applications.

Author Feedback
Author rebuttal: R2: Is the node weighted version reasonable? What is the consequence for the corresponding embedding?

A: Node weighted theta, see (7), is not new, but was proposed in [17]. Its kernel characterisation is new. The edge weighted version is new and is the main focus of the paper. In the combinatorial optimization setting, the node weighted version gives an upper bound on the weight of a max weight independent set as shown in (9) (and in [17]). Geometrically, it looks for orthonormal embeddings which have different pre-specified norms. In terms of theta, Knuth [17] observes that 0-weight nodes might as well be absent from the graph.

R2: What is the impact of the extension to edge-weighted graphs in (10)?

A: The embedding, instead of orthonormal constraints as in the original definition, now imposes constraints specified by the similarity matrix i.e. the cosine similarity of the vectors assigned to two vertices cannot exceed the corresponding entry of the similarity matrix

R2: It is not clear to me what values the kernel K is allowed to take.

A: Note in (3) K_{ij}=0 only when (i,j) do not have an edge. Otherwise it could be anything, subject to K being psd. This will be clarified

R1: The authors suggest using vertices with the largest alphas as initial centroids. ... In the SVM setting these points are typically outliers ... and don't represent the bulk of the data. I understand that the authors want 'diversity', but the explanation in lines 263-265 seems to be at odds with the stated goal of finding good centroids in 4.1. Do we want representative points or influential points?

A: This is an interesting question which we haven't answered fully yet. We argue that theta can be seen as a measure of diversity. An interesting property is that theta will also try to align most of the labellings, u_i, with the labellings of the support vectors and hence the notion of centroids. We also note that the selected vectors are used for _initializating_ k-means, which is why diversity is important. If there is a problem with outliers, one option is to use a soft-margin SVM. Last, in many graphs, such as social networks, nodes are associated with an activity level. This can be used as a node weight to trade off diversity for 'representativeness'

R1: To recover the embedding, the authors propose to use a truncated SVD. ... How much better than a full-rank representation can we do and still have good performance? The authors choose \sqrt(n) for the truncation order, but it would be useful to have some intuition as to the impact

A: Truncation was used only for MaxCut. The order is supported by Goemans & Williamson [10] who showed that sqrt(2n) is sufficient. For clustering, the full decomposition was used. This will be clarified. As R1 suggests, the efficiency depends on the structure of the graph, and how the embedding is used. Intuitively, the more clustered the graph, the closer the necessary dimensionality would be to the number of clusters. This is an interesting question that could be addressed in future work

R2: There is no comparison with the unweighted version for binarized graphs. How do I know that the weights contributed to the solution?

A: This comparison was left out as it is not clear how to binarize the graphs. In the MaxCut case, weights are {-1,0,1}. In the clustering case, weights are continuous, and there is no obvious way to threshold them. Thresholding at e.g. 0.5, might have drastic effects as previously relatively unconstrained vector pairs (cosine sim < 0.5) are now constrained to be orthogonal (sim=0).

R1 & 2: In the experiments on maxcut (Tbl 2), the authors have selected 6 graphs from the Helmberg & Rendl set of 24 weighted graphs. Why?

A: The reason for selecting these 6 graphs was because the published results of the SDP method [7] and the scatter search [21] (as well as Festa, 2002) contain only these graphs (out of the weighted).

R4: The introduced weighted Lovasz's number has to be related to the one in "Weighted Laplacians and the Sigma Function of a Graph" by Chung & Richardson.

A: We are aware of Chung & Richardson's paper. Although it looks superficially relevant to our problem, it is in fact addressing the unweighted graph case, and the sigma function they discuss is the same as the Delsarte version of the theta function for UNWEIGHTED graphs (5), as they note in the paper.

R2: [Regarding issues with notation]

A: The notation of the introductory definitions will be improved in the final version.

R2: The definition of E with respect to the graph G

A: G = (V,E) is mentioned before (2). We will add it before (1).

R2: The dimensionality of the space in which the u_i live

A: Theta optimizes over all labellings over all dimensions, hence dimension is not mentioned

R2: The definition of the K appearing from equation (2)

A: K is defined in (3)

R2: [Using \forall instead of comma]

A: This will be fixed for the final version.